# Excessive Facial Wrinkling Is Associated with COPD Occurrence—Does COPD Damage Skin Beauty and Quality?

**DOI:** 10.3390/ijerph20031991

**Published:** 2023-01-21

**Authors:** Jan Szczegielniak, Katarzyna Bogacz, Jacek Łuniewski, Marcin Krajczy, Wiesław Pilis, Edyta Majorczyk

**Affiliations:** 1Department of Physiotherapy, Faculty of Physical Education and Physiotherapy, Opole University of Technology, Proszkowska Street 76, 45-758 Opole, Poland; 2Department of Health Sciences and Physiotherapy, Faculty of Health Sciences, Jan Dlugosz University in Czestochowa, Armii Krajowej Street 13/15, 42-200 Częstochowa, Poland

**Keywords:** COPD, facial wrinkles, smoking, skin beauty, skin quality

## Abstract

The aim of this study was to investigate facial wrinkling in COPD patients, its relationship with lung function parameters, and the differences in wrinkling between COPD patients and smokers without COPD. The study included 56 patients with COPD with smoking history and 84 controls. Wrinkle intensity was measured and classified using Daniell’s grading system, and the total length of wrinkles was also estimated. The predominant grades of Daniell’s scale were IV–V for COPD patients (89.3% of current and 75.0% of former smokers), III–V for controls who currently smoke (89.2%), and II–III for former (92.9%) and never smokers (100%) controls. These distributions were statistically significantly different, but current and former smokers with COPD and COPD former smokers and control current smokers did not differ. In terms of the total length of wrinkles, the COPD patients possessed significantly longer wrinkles than the control subgroups (all *p*-values were <0.004). Negative correlations between wrinkle length and lung parameters were found. This phenomenon seems to be independent of smoking, but the length of wrinkles is related to lung function parameters. It seems that not only smoking but also COPD damages skin beauty and quality.

## 1. Introduction

Chronic obstructive pulmonary disease (COPD) is defined as a multifactorial disease with gene–environment interactions, leading to airflow limitation through the respiratory tract that is not fully reversible. The worldwide prevalence of COPD is estimated to be 10% in people aged 40 years and older [1]. The two major symptoms of COPD are chronic bronchitis and pulmonary emphysema [2], but many patients have more than one comorbidity, among which facial wrinkling is frequently prevalent [3] and is potentially associated with cigarette smoke exposure (smoking is a major environmental risk factor for COPD development). This phenomenon is called “smoker’s face” and was described for the first time in the mid-19th century [4]. Cigarette smoking negatively affects the skin and is typically associated with skin/facial alterations, such as prominent wrinkles, gauntness of facial features, skin with gray appearance, and swelling of the face [4,5]. In turn, fewer effects are observed for tobacco heating products and electronic cigarettes, which result in skin staining or activation of biomarkers for oxidative stress [6]. This suggests a predominant role of tobacco smoke in generating mild skin inflammation and generally inducing precocious skin aging, with impaired skin beauty and quality manifested by excessive wrinkling, for example [7]. Moreover, it is suggested that smoking-related wrinkle intensity correlates with the level of decline in lung function [8].

Hence, the aim of this study was to investigate facial wrinkling in COPD patients and its relationship with lung function parameters. Additionally, for analysis of the impact of COPD on skin quality damage, we analyzed facial wrinkling in cases with COPD in comparison to healthy controls with a similar smoking history. Moreover, to the best of our knowledge, we are the first to analyze the total length of facial wrinkles in addition to Daniell’s scale.

## 2. Materials and Methods

### 2.1. Study Population

Fifty-six patients (37 males and 19 females) with COPD were enrolled in this study. COPD diagnosis was performed according to the GOLD (Global Initiative for Chronic Obstructive Lung Disease) recommendations and included spirometric analysis with a lung function test performed twice: Before and 15–20 min after bronchodilator application (400 µg of salbutamol). Exclusion criteria were history of bronchial asthma and body mass index (BMI) greater than 25 kg/m^2^. All patients were characterized by a 20–30-year history of smoking, and among the patients, 28 cases were classified as current smokers and 28 as former smokers with at least a 10-year non-smoking period. All patients were recruited from the inpatient and outpatient populations of the Specialized Hospital of the Ministry of Internal Affairs and Administration (MSWiA) in Głuchołazy.

For the healthy control group, 84 volunteers (56 men and 28 women) were recruited. None of the subjects had a COPD diagnosis or any other lung disease and had a BMI lower than 25 kg/m^2^. The controls were spirometrically tested once for lung function. Among the controls, 56 individuals were defined as smokers (28 current and 28 former with the same criteria as the COPD patients), and 28 participants were never smokers. 

To eliminate the influence of UV exposure in both research groups, the exclusion criteria included outdoor work without sun protection and/or frequent holidays in a sunny climate without sun protection (data were obtained using a questionnaire dedicated to analysis of exposure to sunlight). 

All the members of the patient and control groups were of Polish Caucasian ethnicity, and detailed characteristics are shown in Table 1. The experiment was approved by the Ethics Committee at the Municipal Hospital in Nysa (No. 4/2018) in accordance with the Declaration of Helsinki. Signed informed consent was obtained from all persons tested. 

### 2.2. Wrinkle Analysis

The degree of wrinkling was determined on the left and right sides of the face. Each patient was independently assessed by two clinicians, who were blinded to the smoking status and spirometry results. The range of facial wrinkles was measured using Daniell’s six-point scale. Daniell’s scale is a reliable tool used to rate various degrees of wrinkling: Grade I: Essentially unwrinkled—two or three shallow wrinkles, usually less than 0.5 cm in length, may be present in each crow’s foot area, grade II: Several wrinkles, each of which may be 3 cm long—the number of significant wrinkles on each side may be between two and six, grade III: Several prominent wrinkles on each side, 3 to 4 cm long—many smaller wrinkles may be present as well, while increased wrinkling may be present in the forehead skin, but little wrinkling in the cheek areas, grade IV: Wrinkles extend from the crow’s foot area superiorly and inferiorly, usually 5 cm or more—if wrinkles are of unusual depth, they may be 4 cm long, and wrinkles extend over the cheek areas (zygomatic ridge). Men in this grade frequently exhibit prominent wrinkling of the forehead and posterior nuchal region, grade V: Wrinkles extend from the crow’s foot area and are prominent over the cheeks and forehead, grade VI: Profound wrinkling extending over most of the face [5]. 

Moreover, the total length of wrinkles was estimated by the summed length of all wrinkles.

### 2.3. Statistical Analysis

For anthropometric and spirometry parameters, as well as total wrinkle length, the Shapiro–Wilk method was used for testing the normality of the samples. The intragroup differences were analyzed using one-way analysis of variance (ANOVA), and appropriate Tukey post-hoc tests were applied when significant interactions in ANOVA were identified. For correlation between total wrinkle length and the other parameters, Spearman’s rank correlation coefficient test was performed. For correlation between Daniell’s scale grades and lung function parameters, the Kruskal–Wallis H statistic with post-hoc Dunn’s test was used. These statistical analyses were performed with Statistica v.12 (StatSoft, Inc., Tulsa, OK, USA). Intragroup differences in gender and Daniell’s score were estimated using the chi-square test (2 × 5 tables) and the Vassarstats website for statistical analysis (http://vassarstats.net/newcs.html, accessed on 5 December 2022). As a correction for multiple comparisons, a Bonferroni formula (multiplying the *p*-value with the number of statistical tests) was used when *p*-values were significant. 

*p*-values ≤ 0.05 were considered significant.

## 3. Results

The analyses were performed on a group of 56 patients with COPD (divided into two subgroups: Current and former smokers) and 84 age-, sex-, BMI-, and sun exposure-matched controls without COPD who were classified as current, former, and never smokers. The COPD patients from both subgroups were characterized by significantly lower lung function parameters (FEV1% and FEV1/FVC) than the controls (Table 1).

For wrinkle intensity analysis, when the frequencies of Daniell’s scale grading were compared, significant differences were found (Table 2). None of the analyzed cases showed grade I of Daniell’s scale. In the group of current smokers with COPD, the grading of wrinkles was: Grade III in 3.6% of patients, grade IV in 39.3% of patients, grade V in 50.0% of patients, and grade VI in 7.1% of patients. This distribution did not differ from that of the group of former smokers with COPD (III in 17.9%, IV in 28.6%, V in 46.4%, and VI in 7.1% of patients; *p* = 0.32). On the contrary, significant differences were found when compared to all control groups (*p* = 0.05, *p* < 0.001, and *p* < 0.001 with current smokers, former smokers and never smokers, respectively; Table 2). In turn, the COPD group of former smokers did not differ from the current smoker controls (*p* = 0.10), but differences in comparison to the former smokers of the control group were noted (*p* < 0.001; Table 2). Moreover, the distributions of Daniell’s grades were significantly different in the three control groups (*p* < 0.05 for all comparisons). The never smoker control group was characterized by grades II (85.7%) and III (14.7%), but in the current as well as former smokers without COPD, grades IV (21.4% and 3.6%, respectively) and V (32.1% and 3.6%, respectively) were also observed (Table 2).

The wrinkle intensity was also analyzed using the total length of wrinkles that was different in the analyzed group (ANOVA result showed *p* < 0.00001). The longest wrinkles were observed in COPD patients who currently smoke (528.1 ± 83.4 mm), and this mean length was similar to data from the COPD patients with cigarette cessation (507.4 ± 91.2 mm; *p* = 0.92). Both COPD patient subgroups were characterized by significantly longer wrinkles than all control groups (all *p*-values were <0.004), with differences among the control groups also being observed (total length of wrinkles was 421.3 ± 131.2 mm in current smokers, 315.0 ± 80.3 mm in former smokers, and 209.7 ± 61.2 mm in never smokers; Figure 1).

Moreover, relationships between total wrinkle length and lung function parameters, as well as anthropometrical data, were calculated. There were no significant relationships between total wrinkle length and age and BMI. However, as shown in Figure 2, there were negative correlations between the length of wrinkles and FEV1 and FEV1/FVC (% of predicted), with r-values estimated as −0.77 and −0.58 (for FEV1% and FEV1%/FVC, respectively). 

In addition, the total wrinkle length significantly corresponded with wrinkle intensity, identified using Daniell’s scale (H = 419.82, *p* < 0.001). Then, the relationships between Daniell’s scale grades and FEV1 and FEV1/FVC (% of predicted) were also analyzed, and statistically significant relationships were observed, with *p* < 0.001 and H statistics of −209.16 and −211.02 (for FEV1% and FEV1/FVC%, respectively).

## 4. Discussion

In this study, we analyzed COPD-related impairment of skin quality via wrinkles, measured using two methods: Daniell’s grading and total length of wrinkles. We found that COPD patients who currently smoke and former smokers with COPD were characterized by a similar intensity of facial wrinkles. Both COPD subgroups possessed more intensive wrinkling than smokers without the disease. Nevertheless, significant differences between control subgroups were observed, and current smokers had the most intense wrinkles, but the controls who had never smoked were characterized by mild wrinkling. 

It has been known for more than 150 years that smoking causes alterations of facial skin and features, identified as “smoker’s face” [3,4]. Its symptoms include predominant wrinkling [3,7] and seem to be associated with both beauty damage and life quality impairment. It has been recognized that a 40-year-old smoker resembles a 70-year-old non-smoker in terms of wrinkle patterns [9]. It is known that individuals who smoke are more likely (approximately 4.7 times) to have wrinkles than never smokers [10,11]. 

Moreover, tobacco smoke exposure is a major factor in the development of COPD [1,8]. In turn, it seems more likely that patients affected with COPD exhibit “smoker’s face” due to smoking history. Our analysis showed that current and former smokers with COPD did not differ in the level of wrinkling, but both subgroups possessed more wrinkling than the corresponding control groups. In our opinion, this finding may be recognized as demonstrating that not only smoking but also COPD occurrence affects excessive face wrinkling. It is considered that smoking, as an element of air pollution, correlates with signs of skin aging but also negatively affects the cardiovascular and respiratory systems [12]. Thus, our results suggest that it cannot be excluded that COPD-related “smoker’s face” is a result of both smoking and COPD course in either a multiplicative or additive manner. Unfortunately, we were unable to recruit a group of COPD patients who had never smoked, which would have been helpful in considering COPD as a factor of excessive wrinkling. To date, it is mostly considered that environmental factors induce extrinsic skin aging and impaired skin beauty and quality [13], and this process is especially stimulated by sun exposure (UV radiation), as well as tobacco smoke exposure [14,15]. To eliminate the influence of UV exposure, one of the exclusion criteria was outdoor work without sun protection and/or frequent holidays in a sunny climate without sun protection. These data were obtained from a questionnaire and thus may not be fully true, constituting one of the limitations of our work. Nevertheless, our results confirm the finding of tobacco smoke influencing skin aging: The control group of never smokers was characterized by the mildest wrinkling process. All the subjects from this group had a wrinkle grade of ≤2 in accordance with Daniell’s scale, and a mean of total wrinkle length of 209.7 ± 61.2 mm. These results were significantly lower in comparison to control smokers (current and former). Interestingly, significant differences were found when control current and former smokers were compared, suggesting that quitting smoking may protect against expanded wrinkling. On the contrary, all control groups were characterized by lower wrinkling intensity than COPD patients, who did not differ in the context of smoking status (current vs. former). 

Our results do not contradict those of Patel and coworkers, who documented the relationship between nicotine abuse, wrinkling, and airway obstruction [8]. In accordance with this, our results suggest a negative correlation between the total length of wrinkles and lung function parameters. The patients with lower FEV1% and FEV1/FVC showed longer face wrinkles. Similar findings were described in the SALIA cohort study [16]. This phenomenon could be explained at the molecular level: Both organs (lung and skin) possess an extracellular matrix (ECM) composed of collagen and non-collagen elements (e.g., elastin and proteoglycans) [17]. ECM components are degraded by proteolytic enzymes, including matrix metalloproteinases (MMPs) [18]. This family of enzymes is recognized as a key factor in lung ECM turnover and the members of the family, including MMP-9 and MMP-12, seem to be involved in an imbalance in protease and antiprotease activity, leading to the development of emphysema [19,20,21]. On the contrary, in smokers’ skin, decreased levels of collagen I and collagen III were observed, and this finding may be a result of the imbalance (decreased collagen biosynthesis and/or increased degradation) related to the overexpression of MMPs [14,22,23]. Interestingly, all enzymes seem to be induced by tobacco smoke and other air pollution, factors involved in both processes of skin quality impairment and lung emphysema.

Markers of lung function decline have been observed in the skin [24]. However, with reference to current knowledge, we agree with the conclusion of Zouboulisa and Makrantonakib, who stated that it is necessary to analyze skin wrinkling in the absence of airway obstruction and the latter in the absence of signs of extrinsic aging in order to prove skin wrinkling as a specific predictor of airway obstruction diseases [24]. We postulate that analysis of COPD patients without smoking history would allow considering COPD as a potential factor of skin beauty damage via wrinkling extension. The lack of a non-smoking COPD cohort does not allow excluding either physiological or molecular effects of smoking on skin damage. Therefore, it seems to be a key limitation of our study for analysis of the isolated influence of COPD development on wrinkling.

On the contrary, our results confirm the occurrence of extensive wrinkles in COPD patients, who are affected by not only clinical problems but also cosmetic defects. A few cosmetic formulations, especially containing UV filters and antioxidant substances that may reduce visible wrinkles are known [25,26]. Nevertheless, for COPD patients (10% of the worldwide population), it is important to develop cosmetic products and tools dedicated to the effective reduction of COPD-related wrinkles.

Some limitations were identified in our study: As mentioned above, the absence of a non-smoking COPD patient group and subjective analysis of UV light influence on skin aging. Nevertheless, the broader clinical analysis of COPD patients, including gas transfer measurement (diffusion lung capacity for carbon monoxide, DLCO) or emphysema in addition to spirometry, as well as comorbidity (e.g., diabetes mellitus, heart failure, and osteoporosis) occurrence, is expected in further research. 

## 5. Conclusions

Our results showed that patients with COPD are characterized by increased wrinkling in comparison to controls with a similar smoking history. The length and Daniell’s scale grades of wrinkles are related to lung function parameters. It seems that not only smoking but also COPD has an impact on skin quality damage. 

## Figures and Tables

**Figure 1 ijerph-20-01991-f001:**
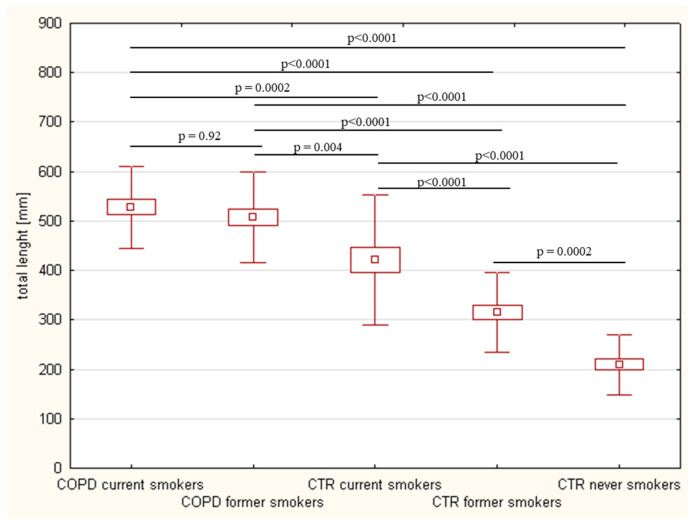
Total length of the wrinkles in the analyzed groups.

**Figure 2 ijerph-20-01991-f002:**
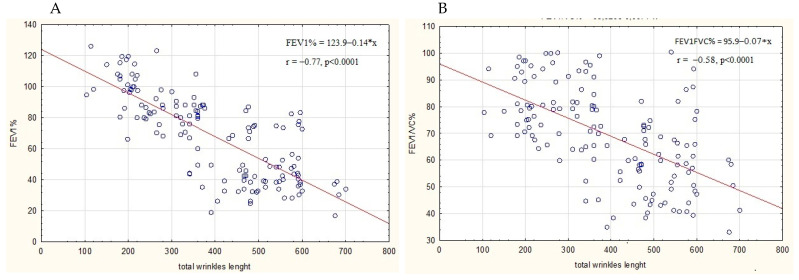
Relationships between total wrinkle length and lung function parameters: (**A**) FEV1% and (**B**) FEV1/FVC.

**Table 1 ijerph-20-01991-t001:** Characteristics of the analyzed groups.

Groups (*N*)	Age (Years)	Sex (Women/Men)	BMI (kg/m^2^)	FEV_1_ (% Predicted)	FEV_1_/VC (% Predicted)
COPD current smokers (28)	59.7 ± 5.2	11/17	22.8 ± 2.0	39.8 ± 8.2 *	51.6 ± 9.6 *
COPD former smokers (28)	56.8 ± 6.6	8/20	23.4 ± 1.6	38.7 ± 10.0 *	52.0 ± 9.5 *
CTR current smokers (28)	55.8 ± 4.4	9/19	22.8 ± 1.7	73.5 ± 8.8	78.5 ± 11.9
CTR former smokers (28)	56.5 ± 6.0	6/22	23.1 ± 2.2	85.2 ± 6.2	80.9 ± 11.9
CTR never smokers (28)	59.8 ± 5.2	13/15	23.2 ± 2.0	104.9 ± 11.5	82.7 ± 9.5

*N*, numbers of individuals; COPD, chronic obstructive pulmonary disease groups; CTR, control group; BMI, body mass index; FEV1, forced expiratory volume in 1 s; FEV1%/VC, Tiffeneau index—forced expiratory volume in 1 s % of vital capacity; * significant differences in comparison to all CTR groups (*p* < 0.0001).

**Table 2 ijerph-20-01991-t002:** Intergroup differences in facial wrinkles using classification by Daniell’s six-point scale.

Daniell’s Scale Grades	Groups Frequency [%] and (Number of Positive)				
	COPD Current Smokers	COPD Former Smokers	CTR Current Smokers	CTR Former Smokers	CTR Never Smokers
I	0 (0)	0 (0)	0 (0)	0 (0)	0 (0)
II	0 (0)	0 (0)	10.7 (3)	42.9 (12)	85.7 (24)
III	3.6 (1)	17.9 (5)	35.7 (10)	50.0 (14)	14.3 (4)
IV	39.3 (11)	28.6 (8)	21.4 (6)	3.6 (1)	0 (0)
V	50.0 (14)	46.4 (13)	32.1 (9)	3.6 (1)	0 (0)
VI	7.1 (2)	7.1 (2)	0 (0)	0 (0)	0 (0)
Statistical analyses					
**Comparison**			*p*-Value/P_corr_		
COPD current smokers vs. COPD former smokers			0.32		
COPD current smokers vs. CTR current smokers			0.005/0.05		
COPD current smokers vs. CTR former smokers			<0.0001/<0.001		
COPD current smokers vs. CTR never smokers			<0.0001/<0.001		
COPD former smokers vs. CTR current smokers			0.10		
COPD former smokers vs. CTR former smokers			<0.0001/<0.001		
COPD former smokers vs. CTR never smokers			<0.0001/<0.001		
CTR current smokers vs. CTR former smokers			0.0007/0.007		
CTR current smokers vs. CTR never smokers			<0.0001/<0.001		
CTR former smokers vs. CTR never smokers			0.002/0.02		

## Data Availability

Data available on request due to privacy.

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
