# Peer review of "Excessive Facial Wrinkling Is Associated with COPD Occurrence—Does COPD Damage Skin Beauty and Quality?"

_ijerph, 2023, doi:10.3390/ijerph20031991_

Round 1
Reviewer 1 Report
The present studz is well designed and conducted, however I could suggest to transform some tables in images to be more clear and to valorize the results obtained so far. For example, the table 3 could be a good image with xy graphics regarding the correlations obtained in this study.
I could also suggest one paragraph regarding the importance of these findings for the cosmetic industry. Please include also the incidence of the present disease in this discussion. It seems a very important find for the claim validation of cosmetic products which are able to reduce the visible wrinkles. Please also improve the conclusion...these finds maz impact the selection of participants for cosmetic evaluation for example. I will suggest some literature.
Cipriani, E., Bernardi, S., & Continenza, M. A. (2016). Wrinkles: origins and treatments. Advances in Cosmetics and Dermatology, 2(1), 1-7.
Gianeti, M. D., & Maia Campos, P. M. (2014). Efficacy evaluation of a multifunctional cosmetic formulation: the benefits of a combination of active antioxidant substances. Molecules, 19(11), 18268-18282.
Özkoca, D., Aşkın, Ö., & Engin, B. (2021). Treatment of periorbital and perioral wrinkles with fractional Er: YAG laser: What are the effects of age, smoking, and Glogau stage?. Journal of Cosmetic Dermatology, 20(9), 2800-2804.
Langton, A. K., Tsoureli‐Nikita, E., Merrick, H., Zhao, X., Antoniou, C., Stratigos, A., ... & Griffiths, C. E. (2020). The systemic influence of chronic smoking on skin structure and mechanical function. The Journal of Pathology, 251(4), 420-428.
Araco, A., & Francesco, A. (2021). Prospective randomized clinical study of a new topical formulation for face wrinkle reduction and dermal regeneration. Journal of Cosmetic Dermatology, 20(9), 2832-2840.
Zargaran, D., Zoller, F., Zargaran, A., Weyrich, T., & Mosahebi, A. (2022). Facial Skin Aging: key concepts and overview of processes. International Journal of Cosmetic Science.
Author Response
The present study is well designed and conducted, however I could suggest to transform some tables in images to be more clear and to valorize the results obtained so far. For example, the table 3 could be a good image with xy graphics regarding the correlations obtained in this study.
Our answer: thank you for this suggestion, we transform tables 3 to image (Figure 2).
I could also suggest one paragraph regarding the importance of these findings for the cosmetic industry. Please include also the incidence of the present disease in this discussion. It seems a very important find for the claim validation of cosmetic products which are able to reduce the visible wrinkles. Please also improve the conclusion...these finds may impact the selection of participants for cosmetic evaluation for example. I will suggest some literature.
Our answer: thank you for this suggestion. Now, we added recommended elements, and we revised the conclusion.
Cipriani, E., Bernardi, S., & Continenza, M. A. (2016). Wrinkles: origins and treatments. Advances in Cosmetics and Dermatology, 2(1), 1-7.
Gianeti, M. D., & Maia Campos, P. M. (2014). Efficacy evaluation of a multifunctional cosmetic formulation: the benefits of a combination of active antioxidant substances. Molecules, 19(11), 18268-18282.
Özkoca, D., Aşkın, Ö., & Engin, B. (2021). Treatment of periorbital and perioral wrinkles with fractional Er: YAG laser: What are the effects of age, smoking, and Glogau stage?. Journal of Cosmetic Dermatology, 20(9), 2800-2804.
Langton, A. K., Tsoureli‐Nikita, E., Merrick, H., Zhao, X., Antoniou, C., Stratigos, A., ... & Griffiths, C. E. (2020). The systemic influence of chronic smoking on skin structure and mechanical function. The Journal of Pathology, 251(4), 420-428.
Araco, A., & Francesco, A. (2021). Prospective randomized clinical study of a new topical formulation for face wrinkle reduction and dermal regeneration. Journal of Cosmetic Dermatology, 20(9), 2832-2840.
Zargaran, D., Zoller, F., Zargaran, A., Weyrich, T., & Mosahebi, A. (2022). Facial Skin Aging: key concepts and overview of processes. International Journal of Cosmetic Science.
Reviewer 2 Report
In the manuscript “Excessive facial wrinkling is associated with COPD occurrence – does COPD damage skin beauty and quality?”, the authors investigate the correlation between COPD and wrinkle formation in the presence or absence of smoking. It is interesting to speculate that the impairment of lung function would be related to skin damage.
However, in its present state the manuscript fails to address some key concerns –
1. The absence of a non-smoking COPD cohort makes it difficult to rule out the effects of smoking to wrinkle development. Although the former smokers have a 10 year non-smoking period, there is a possibility that the physiological/molecular effects of smoking prevailed as evidenced by the development of COPD.
2. The authors should test the presence of other facial alterations such as skin colour, health etc. to further confirm the degree of influence of COPD on skin health.
3. Within each group, there seems to be a distribution of Daniell’s scale grades, however, to calculate statistics the average results have been considered. It would be good to show how the wrinkle intensity distribution correlates with the wrinkle length in both the control and COPD groups? Within the COPD groups, do the different cohorts of wrinkle intensity (Grade III, IV, V and VI) also correlate with increasing severity of lung function impairment?
4. What do the comparisons within each grade of wrinkle intensity (eg. Grade V) across both COPD groups and smokers show with respect to wrinkle length? Based on this result, what would the authors comment on the effects of smoking as compared to just COPD?
5. It is interesting that there is no relationship between wrinkle length and age, even in the non-smoking controls. Can the authors indicate the age range of patients tested and comment on the validity of doing this measurement?
Author Response
In the manuscript “Excessive facial wrinkling is associated with COPD occurrence – does COPD damage skin beauty and quality?”, the authors investigate the correlation between COPD and wrinkle formation in the presence or absence of smoking. It is interesting to speculate that the impairment of lung function would be related to skin damage.
However, in its present state the manuscript fails to address some key concerns –
- The absence of a non-smoking COPD cohort makes it difficult to rule out the effects of smoking to wrinkle development. Although the former smokers have a 10 year non-smoking period, there is a possibility that the physiological/molecular effects of smoking prevailed as evidenced by the development of COPD.
Our answer: thank you for this comment. You are perfectly right that lack a non-smoking COPD cohort generates some interpretation difficulty identification of actual effects of smoking to wrinkle development. We agree with you that there is a possibility that the physiological/molecular effects of smoking prevailed as evidenced by the development of COPD. This suggestion has been added to the text.
- The authors should test the presence of other facial alterations such as skin colour, health etc. to further confirm the degree of influence of COPD on skin health.
Our answer: thank you for this comment. We agree that extensive analysis of skin condition (wrinkles, colour, health etc) may support to confirm COPD influence on skin beauty and quality. Unfortunately, at the time the experiments were performed, we did not take other data from patients and controls.
- Within each group, there seems to be a distribution of Daniell’s scale grades, however, to calculate statistics the average results have been considered. It would be good to show how the wrinkle intensity distribution correlates with the wrinkle length in both the control and COPD groups? Within the COPD groups, do the different cohorts of wrinkle intensity (Grade III, IV, V and VI) also correlate with increasing severity of lung function impairment?
Our answer: thank you for this suggestion. We correlated the wrinkle length and wrinkles intensity using Kruskal-Wallis H Statistic with post-hoc Dunn's test for test a correlation between a continuous and categorical variable. Moreover, in the same test, we tested correlations between wrinkles intensity and lung function parameters. Additional data have been added to the text.
- What do the comparisons within each grade of wrinkle intensity (eg. Grade V) across both COPD grotups and smokers show with respect to wrinkle length? Based on this result, what would the authors comment on the effects of smoking as compared to just COPD?
Our answer: thank you for this comment. As you suggest, we performed additional analysis using combined groups of current smokers (COPD and controls) as well as former smokers and non-smoking individuals (only controls). We obtained significant result; however, a limitation of this analysis is an absence of non-smoking COPD group. Moreover, our previous analyses showed significant differences: COPD smokers vs. controls smokers (both: current and former). Due to all mentioned findings we concluded as was previously considered: “It seems that not only smoking but also COPD has an impact on skin quality damage. “
- It is interesting that there is no relationship between wrinkle length and age, even in the non-smoking controls. Can the authors indicate the age range of patients tested and comment on the validity of doing this measurement?
Our answer: You are perfectly right that surprisingly we do not shown relationship between age and wrinkles length. In our study, the age range of patients is of 50-69 years. The wrinkles analysis was performed 2 times for each patient (two clinicians, who were blinded to the smoking status and spirometry result). We did not observe significant differences between these 2 measurements. As our result are reproducible, they are valid.
Reviewer 3 Report
This paper is well written with good structure and has some interesting findings. I add the following points that should be revised to enhance the quality of this paper.
Major revisions
1. Relationship between skin abnormality and COPD has already been reported as follows.
Michael E O'Brien et al, Respir Res. 2019 Jun 24;20(1):128. doi: 10.1186/s12931-019-1098-7.
Patel BD, Loo WJ, Tasker AD, et al. Smoking related COPD and facial wrinkling: is there a common susceptibility? Thorax. 2006;61(7):568–571.
Maclay JD, McAllister DA, Rabinovich R, et al. Systemic elastin degradation in chronic obstructive pulmonary disease. Thorax. 2012;67(7):606–612.
Joanna Miłkowska-Dymanowska, Adam J Białas, Anna Zalewska-Janowska, Paweł Górski, Wojciech J Piotrowski. Loss of skin elasticity is associated with pulmonary emphysema, biomarkers of inflammation, and matrix metalloproteinase activity in smokers.
Respir Res. 2019 Jun 24;20(1):128. doi: 10.1186/s12931-019-1098-7.
Authors should perform further examinations to find something new.
I recommend the following analyses.
① Analysis of the severity of facial wrinkling according to mild, moderate, and severe CODPs classified by pulmonary function test
② Analysis of the severity of facial wrinkling according to DLCO or emphysema.
③ Analysis of the severity of facial wrinkling based on the presence or absence of COPD comorbidities including DM, heart failure, and osteoporosis, if possible.
2. Facial wrinkling can function as a marker for skin abnormality, but can be influenced by sunlight exposure as authors wrote in the manuscript. Therefore, that point should be described as one of the limitations in the discussion.
Author Response
This paper is well written with good structure and has some interesting findings. I add the following points that should be revised to enhance the quality of this paper.
Major revisions
- Relationship between skin abnormality and COPD has already been reported as follows.
Michael E O'Brien et al, Respir Res. 2019 Jun 24;20(1):128. doi: 10.1186/s12931-019-1098-7.
Patel BD, Loo WJ, Tasker AD, et al. Smoking related COPD and facial wrinkling: is there a common susceptibility? Thorax. 2006;61(7):568–571.
Maclay JD, McAllister DA, Rabinovich R, et al. Systemic elastin degradation in chronic obstructive pulmonary disease. Thorax. 2012;67(7):606–612.
Joanna Miłkowska-Dymanowska, Adam J Białas, Anna Zalewska-Janowska, Paweł Górski, Wojciech J Piotrowski. Loss of skin elasticity is associated with pulmonary emphysema, biomarkers of inflammation, and matrix metalloproteinase activity in smokers.
Respir Res. 2019 Jun 24;20(1):128. doi: 10.1186/s12931-019-1098-7.
Authors should perform further examinations to find something new.
I recommend the following analyses.
① Analysis of the severity of facial wrinkling according to mild, moderate, and severe CODPs classified by pulmonary function test
Our answer: thank you for the suggestion. All COPD patients were recruited from the inpatient and outpatient populations of the Rehabilitation Department of the Specialized Hospital during performing of pulmonary rehabilitation programme. Due of this, the patients were in severe and very severe stages of COPD (accordingly to GOLD stages: FEV1% <50) with exception of 2 cases in moderate COPD. This clinical characteristic of COPD patients does not allow us to perform suggested analysis.
② Analysis of the severity of facial wrinkling according to DLCO or emphysema.
③ Analysis of the severity of facial wrinkling based on the presence or absence of COPD comorbidities including DM, heart failure, and osteoporosis, if possible.
Our answer: thank you for the comments ② and ③. In our opinion, the extensive analysis using clinical data (DLCO, emphysema, comorbidities occurrence) may be important. Unfortunately, at the time the experiments were performed, we did not take other data from COPD patients. Therefore, providing of these type of analysis needs new study design and we already stated that this task be one of our future works. Thank you.
- Facial wrinkling can function as a marker for skin abnormality, but can be influenced by sunlight exposure as authors wrote in the manuscript. Therefore, that point should be described as one of the limitations in the discussion.
Our answer: thank you for this comment. Suggested aspect has been discussed to the text).
Round 2
Reviewer 2 Report
The authors have answered comments, and apart from a few spelling and language errors, the manuscript reads well.
Author Response
Thank you, our manuscript has been corrected by English Editing Service as you suggested.
Reviewer 3 Report
Minor revision.
1. The authors were not able to perform the analyses I recommended. Hence, they should add those areas for further analyses to the limitation segment of the discussion.
Author Response
Thank you for this comments. In accordance with your suggestion the lack of analyses has been indicated as a limitation of our work at the end of the discussion.